# The allocation of carbon resources in marine capture fisheries

Guangliang Li[1], Weikun Zhang[2], Hailan Qiu[3]*, Chunlan Tan[4], Juanjuan Niu[5]

**1** School of International Economics and International Relations, Liaoning University, Shenyang, Liaoning, China, **2** School of Social and Public Administration, Lingnan Normal University, Zhanjiang, Guangdong, China, **3** School of Economics and Management, Jiangxi Agricultural University, Nanchang, Jiangxi, China, **4** College of Economics and Management, Shanghai Ocean University, Shanghai, China, **5** School of Economics, Liaoning University, Shenyang, Liaoning, China

These authors contributed equally to this work.

* qiuhailan@jxau.edu.cn

**Data Availability Statement:** The data underlying the results presented in the study are available from "China Fishery Statistical Yearbook", and all data in the paper are downloaded on its website https://www.cafs.ac.cn/info/1397/37342.htm.

## Abstract

Marine fishery carbon emissions play a significant role in agricultural carbon emissions, making resource allocation a crucial topic for the overall marine ecological protection. This paper evaluates the dynamic iteration method as a research approach with the factors of resource allocation consisting of value assessment, optimization objective, difference between value assessment and objective, and optimization calculation. The paper selects the shadow price from the Super-SBM model as the judgment function for the goal value, aiming for the fairness criterion. From an equity standpoint, the allocation of carbon resources in marine capture fisheries proves to be unreasonable. The fishery model exhibits an excessive supply of carbon resources, resulting in wastage, while the green fishery model faces a relatively limited supply, with a focus on energy conservation and environmental protection. To address this issue, this paper proposes a new method and discusses the corrective results. This result shows that the stabilization point achieved is a short-term equilibrium rather than a long-term one. By rectifying the social contradiction of profit-oriented approaches, this research provides a fresh perspective for economic studies and applications, particularly in industrial layout and resource utilization optimization.

## 1 Introduction

Environmental problems caused by large-scale greenhouse gas emissions pose a global challenge, and countries are committed to energy conservation and emission reduction as they develop plans to develop low-carbon economies [1]. Capture fisheries are highly dependent on fuel consumption and its production pattern is characterized by high carbon emissions in agricultural production [2, 3]. Meanwhile, the scale of China's fishing operation is the largest in the world, and the study of fishing emission reduction has important representative significance in the field of agricultural emission reduction and marine emission reduction [4, 5].

According to the pollution shelter theory, countries with strict environmental regulations shift pollution-intensive industries to countries with lax environmental regulations [6, 7]. This

**Funding:** This research is supported by the National Social Science Foundation of China (22CGL027), the Youth Project of Humanities and Social Science Research in University of Jiangxi Province (JJ21227), the Science and Technology Research Project of Jiangxi Provincial Department of Education (GJJ210461), General Programs of Humanities and Social Sciences of the Ministry of Education (20YJAZH113), and Guangdong Philosophy and Social Sciences Planning Youth Project (GD22YGL19).

**Competing interests:** NO authors have competing interests

is manifested by the fact that high-income industries or sectors can be willing to pay more money to maintain a clean environment [8]; in low-income regions or sectors, they are willing to earn income by selling environmental resources [9]. In an allocation of environmental resources dominated by the value objective of utility maximization, high-income regions will have a comparative advantage in clean products and export more clean products, while low-income regions will export more pollution-intensive products [10]. According to factor endowment theory, referring to the research of Ragland et al.(2015), a region with relatively abundant physical capital has a comparative advantage in capital-intensive industries [11], and trade liberalization will cause the region to export more pollution-intensive products and produce more pollution.

In addition, developed countries put developing countries at a disadvantage in carbon emissions trading by blocking the corresponding emission reduction technologies or resources [12]. In market trading, developed countries are able to use their comparative advantage in carbon emissions to restrict the economic and social development of developing countries as a way to achieve carbon monopoly [13, 14].

The root of this contradiction is reflected in the unequal economic status, and in the process of free resource flows, economic interests are used as the guiding criteria for resource flows. The resolution of this contradiction requires the unification of goal orientation and resource utilization to achieve resource allocation guided by value objectives. It is clear that equity is a very important topic in the carbon reduction process. How to properly deal with the allocation of carbon resources? It has been the focus of carbon equity topic.

For this reason, many scholars have discussed the issue of carbon allocation equity. Some scholars use the resource allocation model derived from DEA [15], with efficiency and equity as the objectives, and use the operation and solution approach to complete solving the optimal value by adding the relative efficiency evaluation of DEA as the constraint. Lin et al. (2003), Avellar et al.(2005), and Wang et al. (2013) treated the CO2 emission allowance allocation problem as a cost or efficiency planning problem, by adjusting the allocation of input or output quantities among DMUs, thus changing the location of the original efficiency frontier surface [16–18]. The DEA allocation method used by scholars combines efficiency and equity, but scholars do not go further to discuss a more universal allocation scheme. Obviously, the goals of efficiency and fairness do not represent all value evaluation systems. Thus, this paper introduces a new scheme of resource allocation based on value evaluation as an adjustment target.

In human society and natural evolution, the law of motion of things is filtered according to natural rules and adapted to standards through gradual evolution with dynamic flexibility [19, 20]; that is, it can evolve according to changes in natural conditions, and this natural law is also applicable to the economic field. Resource allocation is in pursuit of a particular goal [21, 22], and this adjustment process is extremely similar to the training of neural networks. Therefore, this paper adopts some of the ideas of neural network calculation process, the algorithm studies the change of the result through the small change of the input, and finally gradually approximates the ideal value through the adjustment of the input elements.

Therefore, this paper abstracts this idea and applies it to the field of resource or industry allocation, taking the shadow price under fair governance of rights and responsibilities as the value target, calculating the adjustment weight value by using the difference between individual value and target value, and finally allocating the resources reasonably through the weight division of rights and responsibilities. The second section introduces the shadow price theory of weight equity and the theory of equitable resource allocation; The third section constructs the framework and mathematical derivation of the resource allocation model; The fourth section conducts the analysis and evaluation of the allocation results; The fifth section elaborates the research conclusions and model prospects.

## 2 Theory

### 2.1 Principles of resource evaluation

In environmental economics, environmental resource products are valued in the following ways: the productivity change approach, the cost of disease approach, the human capital approach, and the opportunity cost approach [23, 24]. Shadow price is a kind of opportunity cost, which indicates: the maximum output obtained by continuing to input a certain resource or emitting a pollution product, i.e., the benefit obtained by expanding the resource input, which can also be understood as the strength of the incentive to expand reproduction. Understood from an environmental governance perspective, the maximum benefit obtained from one unit of pollutant or carbon resource input or output. This is a willingness to expand production for the producer, while for the abatement policy perspective, it is a cost of governance of pollution, which discourages the opportunity cost of willingness to emit by charging taxes or compensation. Thus, carbon credits are a potential benefit, and this potential benefit can be seen as a reward or a penalty.

The shadow price can indicate the degree of resource scarcity, the larger the shadow price indicates that the resource is relatively scarce, and conversely the smaller the shadow price indicates that the resource is relatively abundant, and the shadow price of zero indicates that the resource is in surplus [25]. The valuation theory of shadow price method is accepted by many scholars, which conducts a study on carbon emission price using shadow price, and concludes that shadow price has important guiding significance for environmental tax rate and primary market pricing [26–28].

### 2.2 Model step

This paper considered short-term allocation, but not considered the scale effect and technological progress caused by resource changes. The application and meaning of shadow price have been introduced above. Shadow price is a quantitative method of resource evaluation, which is suitable for this study. When we know the potential shadow price target, we only need to gradually adjust the number of factors affecting the shadow price, so that the shadow price approaches the ideal value, and the corresponding factor number is the quantity obtained in this paper. A more accurate expression of this truth can be clarified below. When a certain goal is defined, the trajectory is constantly modified toward the goal, based on the following detailed steps for modeling.

Adapting standards in dynamics is a common phenomenon in nature [29], and this phenomenon exists not only in the natural sphere, but also in social behavior, such as business profit as the value judgment standard of social competition [30, 31], low-profit business practices are eventually abandoned by businessmen who aim at profit maximization, and the law of this phenomenon also applies to the allocation structure of industries or resources.

The decision steps are shown in Fig 1. First, the carbon resource shadow price is measured by the existing resource and production evaluation system. Secondly, the shadow price has been calculated to select the appropriate adjustment target, and the target selection reflects the value pursuit of the economy and society. Under the goal requirement of unified carbon reduction rights and responsibilities, carbon resources should be allocated to meet the needs of enterprises and reduce potential profit loss as much as possible. Therefore, this paper selects the minimum value of unified shadow price as the adjustment target of carbon emission resources in fisheries.

Then, deviations have been calculated between the shadow price and the target value. Obviously, if the difference between an enterprise or an industry and the overall target value of

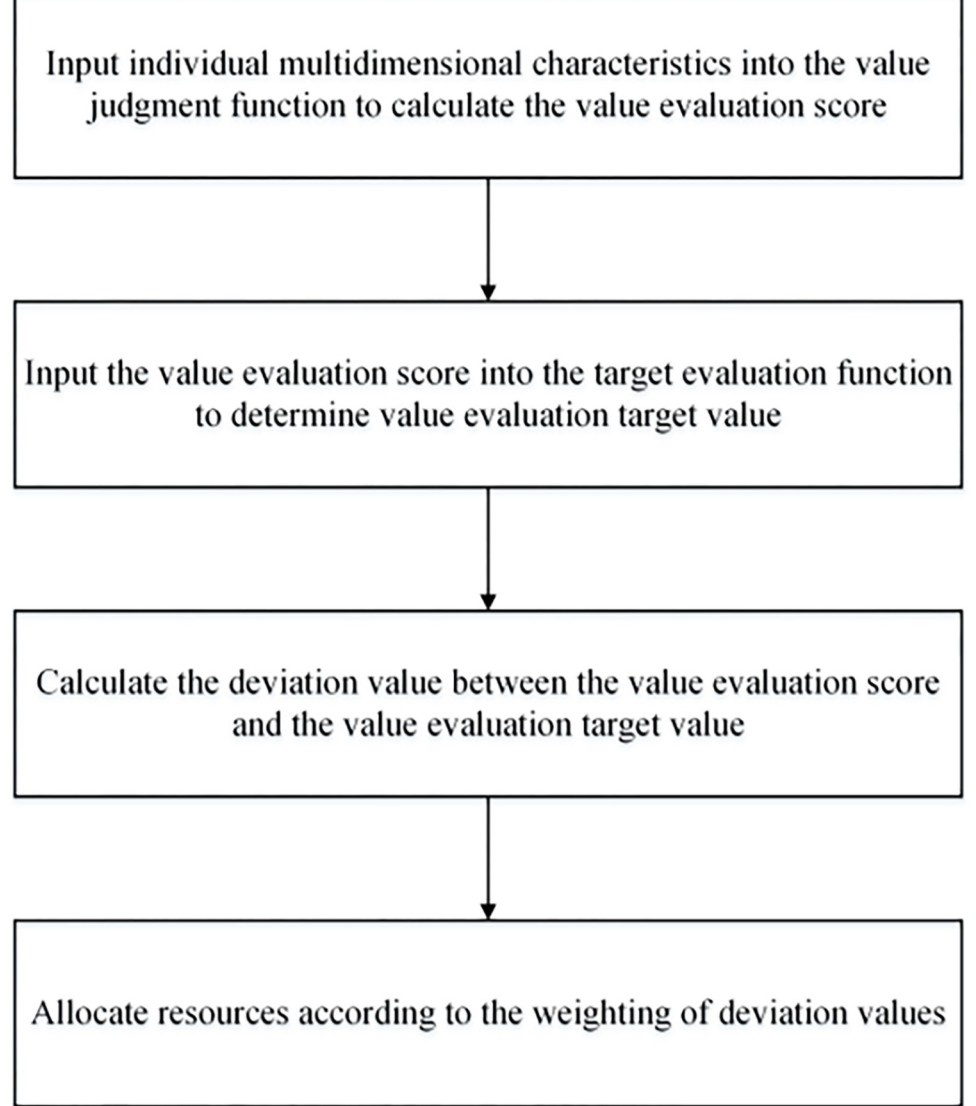

**Fig 1. Summary of steps.**

society is large, i.e., when such shadow price far exceeds the required norms of social expectations, the enterprise or industry should take the initiative to bear a larger responsibility. At this time, the weight value of the number of penal or compensating resources is larger, which satisfies the requirements of social equity. Finally, the weight value is used to update the allocated quantity of resources to narrow the gap between the shadow price and the socially desired shadow price, and after several iterations of the above steps, the resource allocation is finally realized to approach the ideal state.

## 3 Model construction

### 3.1 Data sources

China Fisheries Statistical Yearbook distinguished statistics according to fishing methods. The capture fisheries sector is divided into Trawler, Seine net, Fixed net, Drift net, Angling, and

others, and the first five operating methods are selected as the object of this paper. The data in this paper were obtained from China Fishery Statistical Yearbook, and the carbon emission data are measured by the IPCC carbon emission assessment method.

The number of fishing boats of five fishing methods was selected as the capital variable [32], revealing the capital input. The labor variable was measured by the fishing industry employees, which can directly described the labor input of the fishing industry [33]. Technical variables cannot be directly measured, so this paper used the amount of fishery technology promotion agencies and the amount of fishery technology promotion funds to measure the application of technology [34]. This is a fact that the more technology promotion agencies and technology promotion funds, the faster the technology promotion scope and upgrade are. Technology promotion funds and institutions are a public good and service that can benefit all modes of fishing in the region, so data for one area can represent the input of all fishing methods in the area. Output variables took the production value of capture fisheries (excluding pelagic fishing) as economic output data and carbon dioxide emissions as undesired output [35]. Data was collected from the China Fisheries Statistical Yearbook.

This paper used the usual IPCC emission assessment method to calculate carbon emission data. Carbon emissions from fishing boats are measured as follows [2, 36].

$$T = P \times \rho \times \tau \tag{1}$$

Where $T$ indicates carbon Emissions from Fishing (ton), $P$ indicates Power (kW), $\rho$ indicates Conversion coefficient of fishing vessel fuel consumption (ton/kW), $\tau$ indicates Carbon emission coefficient of diesel, $\tau = 0.5921$. Referring to the data, the calculation method of the Conversion coefficient of fishing vessel fuel consumption is

$$\rho = TDays \times HDay \times 0.000205 (ton/kw/h) \tag{2}$$

Where $TDays$ indicates Total fishing days per year, $Hday$ indicates hours per day. According to official data (Ministry of Agriculture of China), $\rho$ is as follows, Trawler (0.4800 t/kW); Seine net (0.4920 t/kW); Drift net (0.4510 t/kW); Fixed net (0.3280 t/kW); Angling (0.3280 t/kW); Others (0.3120 t/kW). The final numerical results were determined by counting the average operating hours of each type of fishing vessel [2].The descriptive statistics of the above variables with data from China Fishery Statistics Yearbook can be accessed in Table 1.

## 3.2 Algorithm model construction

The algorithm model strategy can be summarized in the following flowchart as shown in Fig 2, which evaluates the evaluation score based on individual characteristics through the value evaluation function, and then evaluates the evaluation score using the ideal target value, and then allocates resources or restructures the industry based on the difference between the individual value score and the ideal target value. For changes in external information or resources, the

**Table 1. Descriptive statistics of variables.**

|  | Carbon emissions (tons) | Production (tons) | Number of Fishing Vessels (Vessels) | Fishery workers (people) | Technology promotion agencies | Technology promotion funds (million yuan) | Production value (million yuan) |
|---|---|---|---|---|---|---|---|
| mean | 62300.74 | 204697.74 | 3059.11 | 644654.09 | 533.32 | 9672.04 | 309900.64 |
| std | 99352.21 | 334965.41 | 4813.68 | 473698.74 | 403.13 | 8440.03 | 474588.75 |
| min | 0.97 | 5 | 1 | 5004 | 13 | 144 | 75.1450 |
| max | 599614.36 | 2103225 | 30601 | 1489275 | 1236 | 36422 | 3238732.56 |

Sources from China Fishery Statistics Yearbook

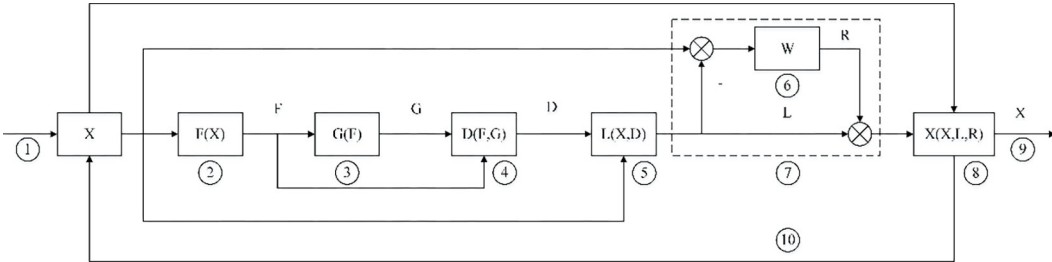

**Fig 2. Algorithm control flow chart.** Note: ① input characteristics; ② value evaluation function; ③ target function; ④ distance function between individual value level and target level; ⑤ distance-based resource adjustment function; ⑥ reallocation weight function; ⑦ total constraint system; ⑧ update resource allocation; ⑨ output results; ⑩ circular feedback.

model can incorporate dynamic functions as simulated fluctuations as a response to information or resource shocks.

The key nodes are described as follows.

①Input characteristics

The variables used for the evaluation are entered with multidimensional characteristics of the individual, such as academic performance in each subject and the number of different asset types in asset management.

②Value evaluation function

Mathematical modeling of value evaluation or natural evaluation criteria, mapping the multidimensional characteristics of individuals into indicator values through value evaluation functions [37, 38], makes individuals quantitatively comparable with each other under certain evaluation criteria. For example, the degree of influence of the type and number of enterprises on regional pollution, the influence of the number of different asset types on risk in asset management, the shadow price of Super-SBM pairwise model [39] is used as the value evaluation function in this paper. The general functional form of the value evaluation function can be expressed as

$$F^t = (f_1^t, f_2^t, \ldots, f_n^t), f_i^t = f(x_i^t) \tag{3}$$

Where $x_i^t$ is the indicator variable or vector of the $i$ individual included in the evaluation system in the $t$ period, $f_i^t$ is the $i$ individual value evaluation index score in the $t$ period, and $F^t$ is the $n$ dimensional score vector composed of the $n$ individual evaluation index scores in the $t$ period. Commonly used evaluation functions are DEA efficiency measures, shadow prices [40], Gini coefficients [41], coupling coordination [42], and other score index models.

③Objective function

The main role of the objective function [43] is to determine the target level, that is, the best state target value that is willing to adjust. The general functional form of the objective function is expressed as

$$G^t = G(F^t) = G(f_1^t, f_2^t, \ldots, f_n^t) \tag{4}$$

Where, $f_i^t$ is the score of the $i$ individual value evaluation function at time $t$ and $G^t$ is the global desirable objective function at time $t$.

The ideal objective function can take the mean expectation value, and the mean expectation value as the objective value means that the overall optimization objective [44] is concentrated

toward the mean and the variance after optimization becomes smaller.

$$G(F^t) = \sum_{i=1}^{n} f_i^t / n \tag{5}$$

From the maximum target value method, this objective enables the overall optimization toward the maximum level value.

$$G(F^t) = Max(f_1^t, f_2^t, \ldots, f_n^t) \tag{6}$$

From the minimum objective value method, this objective enables the overall optimization toward the minimum level value.

$$G(F^t) = Min(f_1^t, f_2^t, \ldots, f_n^t) \tag{7}$$

The constant value method maintains the objective optimized toward a constant level.

$$G(F^t) = C, C \text{ indicates a constant} \tag{8}$$

The above four objective evaluation functions have the same impact on all individuals, i.e., fair impact, and therefore are called global ideal objective functions. After resource allocation based on global ideal objectives, the value evaluation function values of individuals tend to the same standard state. In reality, there will be inconsistencies in the goals pursued by multiple individuals, as well as fluctuations in the target values. To further broaden the hypothesis, based on the differences in the ideal goals pursued by individual characteristics, the ideal goal function can change according to time, so the expanded functional form can be expressed as follows.

$$G_i^t = G(F^t, X_i^t, t) \tag{9}$$

$G_i^t$ is denoted as the objective function of the $i$ individual in the $t$ period, $F^t$ is a vector of $n$ dimensional scores consisting of $n$ individual evaluation index scores for the $t$ period, $X_i^t$ represents the characteristic attributes of $i$ individuals for the $t$ period, and $t$ represents the period.

④Distance function between individual value level and target level

The individual value level and target level distance function can measure the direction and quantity of resource adjustment, and the resource adjustment is carried out by the determined adjustment direction and quantity. The individual value level and target level distance function is defined as

$$D^t = D(F^t, G^t) = (d(f_1^t, G^t), d(f_1^t, G^t), \ldots, d(f_n^t, G^t)) \tag{10}$$

Where $f_i^t$ is the $i$ individual value evaluation function score in the $t$ time period, $G^t$ is the ideal target function value in the $t$ time period. $D^t = (d_1^t, d_2^t, \ldots, d_n^t)$, $d_i^t$ is the distance between the $i$ individual value level and the ideal target value in the $t$ time period, when the individual resource adjustment step $d = d_i^t$. The distance function form is chosen as a monotonic function, and the distance function can be taken as a cubic function in a limitless form as follows.

$$d(f_i^t, G^t) = (f_i^t - G^t)^3 \tag{11}$$

The tanh function, which has a limit form, can also be used [45].

$$d = g(z_i) = \frac{e^{z_i} - e^{-z_i}}{e^{z_i} + e^{-z_i}}, z_i = f_i^t - G^t \tag{12}$$

The advantages and disadvantages of the two are: the cubic distance function is able to approach the ideal value quickly when the gap is large, but the convergence approach speed

becomes slower when approaching the ideal value; the tanh function is able to limit the adjustment range within (-1,1), and the approach speed is slower when the deviation degree is large, and it has higher sensitivity when approaching the ideal value as the slope of the function increases.

⑤ Distance-based resource adjustment function

The resource adjustment function is to allocate resources based on the original resources by using the distance between the individual value level and the target level as a step [46]. The resource adjustment magnitude is controlled by setting an appropriate adjustment ratio $\eta$ to avoid the over-adjustment problem caused by too large a distance function [47, 48]. The resource adjustment function takes the form of

$$L^t = L(X^t, D^t) = (l_1^t, l_2^t, \ldots, l_n^t)^T = (x_1^t \times \eta \times d_1^t, x_2^t \times \eta \times d_2^t, \ldots, x_n^t \times \eta \times d_n^t)$$

$$= (\eta \times d_1^t, \eta \times d_2^t, \ldots, \eta \times d_n^t)(x_1^t, x_2^t, \ldots, x_n^t)^T = M_D^t X^t \tag{13}$$

$$M_D^t = (\eta \times d_1^t, \eta \times d_2^t, \ldots, \eta \times d_n^t) \tag{14}$$

Where $l_i^t$ is the amount of resource adjustment for the $i$ individual in the $t$ period, $\eta$ is the adjustment ratio, and $d_i^t$ is the distance or step between the value level of the $i$ individual in the $t$ period and the desired target value.

The amount of adjustment for the $j$ individual in the $t$ time period is

$$q_j^t = x_j^t \times \eta \times d_j^t \tag{15}$$

The total adjustment is

$$Q^t = \sum_{j=1}^{n} x_j^t \times \eta \times d_j^t \tag{16}$$

1. Where $\eta$ is the adjustment ratio, $\eta$ is characterized as below. Too large a value causes the optimized model to miss the optimal point, too low a value causes the adjustment speed to be slow.

2. ⑥ and ⑦ Redistribution weight function and aggregate constraint system

3. After the above adjustment, the total amount of adjustment $Q^t$ has three states, $Q^t>0$, $Q^t = 0$ and $Q^t<0$, so the resources before and after the adjustment may face the problem of resource surplus or resource shortage. Similarly, the corresponding external resource compensation or resource withdrawal can also be able to bring about corresponding resource changes, and this part of the resources will be represented by $Q_e^t$, so the total amount of resources involved in the reallocation is $Q_c^t = Q^t + Q_e^t$, and this part of the resources can be distributed by the averaging method, and the proportion of the amount of resources obtained by the $i$ individual in the $t$ period is

$$W^t = (w_1^t, w_2^t, \ldots, w_n^t)^T = (\frac{1}{n}, \frac{1}{n}, \ldots, \frac{1}{n})^T \tag{17}$$

The disadvantage of this weight is that the adjusted resource value is negative; to overcome this disadvantage, the model can choose the proportion of individual existing resources to the

overall weight as the redistribution weight, as follows.

$$W^t = (w_1^t, w_2^t, \ldots, w_n^t)^T = (\frac{x_1^t}{\sum_{i=1}^n x_i^t}, \frac{x_2^t}{\sum_{i=1}^n x_i^t}, \cdots, \frac{x_n^t}{\sum_{i=1}^n x_i^t})^T \qquad (18)$$

At this point, the export is as follows:

$$R_c^t = Q_c^t W^t = Q_c^t \times (w_1^t, w_2^t, \ldots, w_n^t)^T = (r_1^t, r_2^t, \ldots, r_n^t)^T = (\frac{Q_c^t x_1^t}{\sum_{i=1}^n x_i^t}, \frac{Q_c^t x_2^t}{\sum_{i=1}^n x_i^t}, \cdots, \frac{Q_c^t x_n^t}{\sum_{i=1}^n x_i^t})^T \quad (19)$$

$R_c^t$ denotes the allocation amount of the resource redundancy or scarcity adjustment under the aggregate constraint, which ensures that the total amount of resources before and after the adjustment remains unchanged.

⑧ Update resource allocation

With the adjustment of the above phases, the new resource values at this point are adjusted as follows

$$X^{t+1} = X^t + L^t - R_c^t \qquad (20)$$

⑨ and ⑩ Output results and loop feedback

When the termination condition is not reached, the value can be readjusted by a circular feedback route ⑩; when the termination iteration condition is reached, the optimization state is terminated and the result is output ⑨. The termination condition can be set as the number of iterations or judged to meet the optimization goal.

In summary, the steps of the mathematical model are synthesized as

Step 1. Value judgment function to calculate individual scores

$$F^t = (f_1^t, f_2^t, \ldots, f_n^t), f_i^t = f(x_i^t)$$

Step 2. Calculation of target values using individual scores

$$G^t = G(F^t) = G(f_1^t, f_2^t, \ldots, f_n^t)$$

Step 3. Calculate the distance function between individual value level and target level

$$D^t = D(F^t, G^t) = (d(f_1^t, G^t), d(f_1^t, G^t), \ldots, d(f_n^t, G^t))$$

Step 4. Resource adjustment using the distance function between individual value level and target level as a step

$$L^t = L(X^t, D^t) = (l_1^t, l_2^t, \ldots, l_n^t)^T$$

Step 5. Total adjustment in accordance with resource constraints

$$R_c^t = Q_c^t W^t = Q_c^t \times (w_1^t, w_2^t, \ldots, w_n^t)^T = (r_1^t, r_2^t, \ldots, r_n^t)^T$$

Step 6. Aligning resources

$$X^{t+1} = X^t + L^t - R_c^t$$

Step 7. Determine whether the termination target is met, if so, terminate the adjustment; if not, enter the next adjustment status.

## 4 Assignment result evaluation

### 4.1 Assessment of carbon resource distribution differences

This paper uses capture fisheries input-output data from 2010–2019 to calculate shadow prices for the output sector, reporting the mean results every two years, as shown in Table 2. Multi-factor ANOVA was used to compare the multidimensional differences in time, operation type, and region, and the results showed that the F-statistics for the time factor was 0.835 (p = 0.584), the F-statistics for the operation type and region factors were 19.400 (p = 0.000) and 18.591 (p = 0.000), and the F-statistic for the cross term for the operation type and region factors was 16.203 (P = 0.000), concluding that there is no difference in the distribution of shadow prices on the time factor and there is a difference in the distribution on the operation type and area factors.

Combined with the analysis of the current development of fisheries fishing in Fig 3, the trawling operation is the largest in China, but the carbon shadow price of trawling operation is at the lowest position, while the economic efficiency of trawling operation is the lowest, which means that the carbon resources of trawling operation are relatively excessive, leading to the waste of resources. The carbon shadow price of Seine operation has a difference between north and south regions, and the carbon shadow price of Liaoning, Tianjin, Hebei, Shandong and Jiangsu is significantly higher than that of Shanghai, Zhejiang, Fujian, Guangdong, Guangxi and Hainan, etc. The amount of carbon resources of northern Seine operation is relatively tighter than that of the south, which indicates the geographical difference of carbon resource

**Table 2. Shadow price data, 2010–2019.**

| Type | Year | Tianjin | Hebei | Liaoning | Shanghai | Jiangsu | Zhejiang | Fujian | Shandong | Guangdong | Guangxi | Hainan |
|---|---|---|---|---|---|---|---|---|---|---|---|---|
| Trawler | 2010–2011 | 0.15472 | 0 | 0.00156 | 0.01092 | 0.20137 | 0 | 0.0145 | 0.01153 | 0 | 0 | 0.00302 |
| Trawler | 2012–2013 | 0.06068 | 0.03777 | 0.01022 | 0.01319 | 0.11469 | 0.00336 | 0.01313 | 0.01411 | 0 | 0.02175 | 0.00279 |
| Trawler | 2014–2015 | 0.13242 | 0.06267 | 0.00965 | 0.00935 | 0.11155 | 0.00332 | 0.0126 | 0.01463 | 0 | 0.02485 | 0.0026 |
| Trawler | 2016–2017 | 0.0818 | 0.05638 | 0.01138 | 0.0088 | 0.11558 | 0.00176 | 0.01236 | 0.01416 | 0 | 0.0278 | 0.00156 |
| Trawler | 2018–2019 | 0.02529 | 0.05052 | 0.00627 | 0.01816 | 0.10526 | 0.00345 | 0.01243 | 0.01443 | 0.01352 | 0.02941 | 0.00271 |
| Seine net | 2010–2011 | 0 | 0 | 0 | 0 | 0 | 0 | 0 | 0 | 0 | 0.10185 | 0 |
| Seine net | 2012–2013 | 0.82853 | 0 | 1.39036 | 0 | 1.42327 | 0.03581 | 0.03587 | 0.28752 | 0 | 0.11851 | 0.00149 |
| Seine net | 2014–2015 | 2.3783 | 1.42353 | 0.4329 | 0 | 2.58297 | 0.03478 | 0.04337 | 0.30761 | 0.08533 | 0.13087 | 0.00237 |
| Seine net | 2016–2017 | 0.34383 | 1.38637 | 0.45672 | 0 | 2.58946 | 0.01701 | 0.02927 | 0.20461 | 0.08203 | 0.10653 | 0.00154 |
| Seine net | 2018–2019 | 0.07297 | 0.24192 | 0.48943 | 0.04636 | 2.35171 | 0.03091 | 0.02235 | 0.16758 | 0.0536 | 0.41725 | 0.00156 |
| Drift net | 2010–2011 | 0.01873 | 0.01414 | 0.00172 | 0.51531 | 0.05719 | 0.0196 | 0.03105 | 0 | 0 | 0.07619 | 0 |
| Drift net | 2012–2013 | 0.04639 | 0.01387 | 0.01132 | 0.27453 | 0.05104 | 0.01688 | 0.02821 | 0.03354 | 0.01094 | 0.1723 | 0.00136 |
| Drift net | 2014–2015 | 0.04199 | 0.01326 | 0.01197 | 0.44377 | 0.04506 | 0.01564 | 0.02694 | 0.06526 | 0.02152 | 0.07573 | 0.00083 |
| Drift net | 2016–2017 | 0.02795 | 0.0154 | 0.01065 | 0.38915 | 0.03748 | 0.01374 | 0.02697 | 0.05933 | 0.02101 | 0.11587 | 0.00068 |
| Drift net | 2018–2019 | 0.00896 | 0.0139 | 0.00747 | 1.14388 | 0.03553 | 0.01132 | 0.02676 | 0.05416 | 0.01914 | 0 | 0.00063 |
| Fixed net | 2010–2011 | 0.26885 | 0 | 0 | 0 | 0.06158 | 0 | 0 | 0 | 0 | 0.99933 | 0.01534 |
| Fixed net | 2012–2013 | 0.57666 | 0 | 0 | 0.14257 | 0.14192 | 0 | 0 | 0 | 0 | 1.00899 | 0.02687 |
| Fixed net | 2014–2015 | 0.4332 | 0 | 0 | 0.29329 | 0.15136 | 0 | 0 | 0 | 0 | 1.09658 | 0.02433 |
| Fixed net | 2016–2017 | 0.31309 | 0 | 0 | 0.27685 | 0.17488 | 0 | 0 | 0 | 0 | 1.11658 | 0.01542 |
| Fixed net | 2018–2019 | 0.08945 | 0 | 0 | 0.58661 | 0.2468 | 0.02655 | 0 | 0 | 0 | 1.35713 | 0 |
| Angling | 2010–2011 | 0.0324 | 3.04718 | 0.03446 | 0.09925 | 49.97289 | 0.05161 | 0.24525 | 0.24339 | 0 | 0 | 0 |
| Angling | 2012–2013 | 0.21584 | 2.89519 | 0.21615 | 0.6913 | 26.60096 | 0.03259 | 0.25597 | 0.12428 | 0.08002 | 0 | 0.01021 |
| Angling | 2014–2015 | 0.19429 | 2.85462 | 0.13425 | 0.08809 | 13.78849 | 0.02692 | 0.24356 | 0.1221 | 0 | 0 | 0 |
| Angling | 2016–2017 | 0.14013 | 2.86634 | 0.10126 | 0.0881 | 17.6864 | 0.02235 | 0.17535 | 0.1437 | 0 | 0 | 0.00255 |
| Angling | 2018–2019 | 0.04004 | 2.47522 | 0.07349 | 0.08902 | 34.27801 | 0.01544 | 0.10534 | 0.1047 | 0.09352 | 0 | 0 |

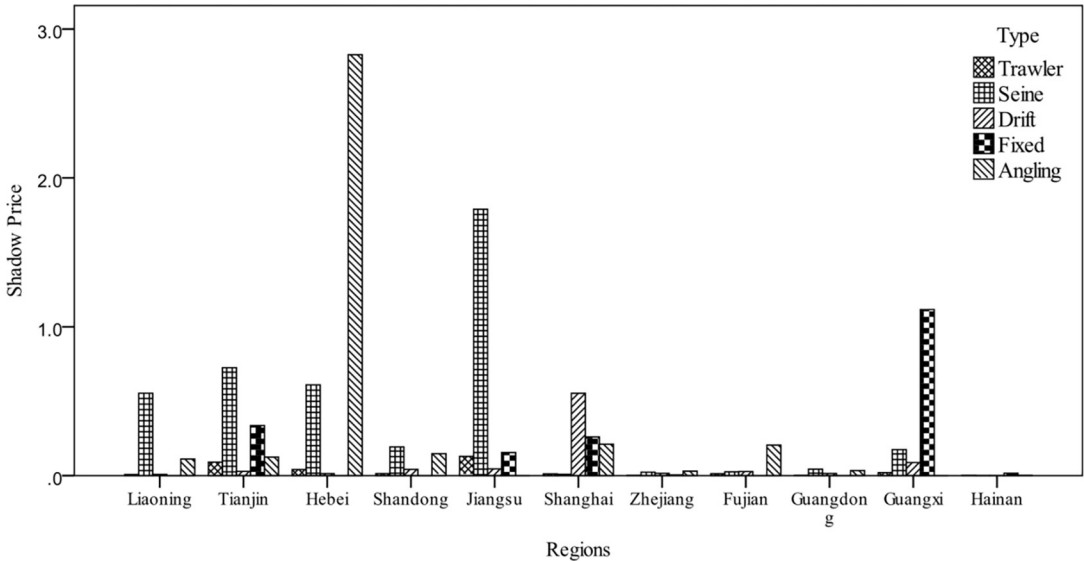

**Fig 3. Average shadow prices, 2010–2019.** Note: The average value of the shadow price of carbon resources for fishing in Jiangsu is 28.43 (the value is too large and not given in the figure).

allocation in capture fisheries. The carbon shadow price is higher in Hebei and Jiangsu Angling industry, which is a low-carbon emission and high-efficiency operation, but due to the low production value and long capital return cycle, which leads to insufficient resource investment, the carbon resources of Angling industry show a scarce state. Summing up the current carbon shadow price allocation pattern, carbon resources have unreasonable resource allocation, carbon resources wasteful fishing mode with sufficient carbon resource supply, energy-saving and environment-friendly green fishing mode with relatively tight carbon resources, and the industrial structure aiming at profit maximization does not meet the value requirement of maximizing carbon resource utilization efficiency. At this time, when using the requirement to assume exactly the same carbon emission reduction responsibility to achieve the suppression of carbon emission reduction, can only suppress the industrial scale of trawlers, and will destroy the more environmentally friendly, low-carbon Angling industry economic output welfare, making the Angling industry allocated to less carbon resources.

Table 2 shows the changes of shadow prices of carbon resources. Due to differences in fishing methods, shadow prices have shown different trends in the past 10 years. In the trend of Trawl shadow price change, Tianjin and Jiangsu showed a trend of decline, while Hebei, Liaoning, Shanghai, Guangdong and Guangxi showed a trend of growth, and Zhejiang, Fujian, Shandong and Hainan showed a relatively stable shadow price. According to the meaning of shadow price, the shadow price approaching 0 means that resources gradually change from shortage to surplus, and the carbon resource prices of Tianjin and Jiangsu decrease, which means that the carbon resources used for trawling in the two places gradually become abundant. The same analysis shows that the growth trend of Hebei, Liaoning, Shanghai, Guangdong and Guangxi indicates that trawl carbon resources are increasingly scarce. In the trend of the shadow price of Seine net, Tianjin, Hebei, Liaoning, Jiangsu, Fujian, Shandong and Guangdong show an increasing and then decreasing trend. Since the shadow price can be used as an indicator of resource scarcity, it can be assumed that the trend of carbon resources in these seven places is changing from gradually scarce to gradually abundant. Shanghai, Zhejiang, Guangxi and Hainan show fluctuating trends.

In the trend of gillnet shadow price changes, Tianjin, Liaoning, Shandong, Guangdong and Hainan show an increasing and then decreasing trend, which indicates that the abundance of carbon resources in these areas is gradually improving. Jiangsu and Zhejiang show a decreasing trend, which indicates that Jiangsu and Zhejiang bear a smaller opportunity cost of carbon resources in the type of Drift net and have more abundant resources, but when carbon resources are more abundant, the pressure to reduce emissions does not have a good effect; on the contrary, Hebei's shadow price decreases and then increases, which means that Hebei bears the pressure to reduce emissions due to the relative shortage of carbon resources.

In the trend of change in the shadow price of Fixed net, Tianjin shows an increasing and then decreasing trend, while Shanghai and Jiangsu show an increasing trend. However, most other regions show a value of 0. Fixed net are characterized as a passive fishing method, which does not rely excessively on fuel consumption, and therefore, most regions are relatively more abundant in carbon resources for Fixed net. In Angling, Zhejiang has been showing a decreasing trend and Guangxi has been at 0, implying that these two regions have higher carbon abundance. Other regions have been more volatile, with no particularly clear trend.

Trawling and Seine net are two fishing methods widely used by fishermen, and the differences in carbon resource sensitivity between the two fishing methods are more obvious in terms of regional distribution, and have more obvious trends such as increasing and decreasing in terms of trend characteristics, which involve less random fluctuations in shadow prices. However, fishing methods that are not commonly utilized, such as Drift net, Fixed net and Angling, exhibit a fluctuating trend. This fluctuating trend is, on the one hand, caused by the lack of inherent stability due to the small scale of the fishing methods, and, on the other hand, the fluctuating trend implies that it is not stable in terms of sensitivity to the pressure to reduce emissions. Therefore, the allocation of carbon resources to this type of fishery should take full account of the production characteristics of small-scale, unstable efficiency and passive fishing. It is clear that this type of fishing needs better policy support for its development, which can hopefully replace the fuel-inefficient and pollutingly destructive fishing methods of trawling.

## 4.2 Results analysis

**4.2.1 Global analysis.** As shown in Fig 4, comparing before and after resource adjustment, the results show that most of the regional sectors need to scale down their carbon emissions under the guidance of the equity goal of the same potential benefits, indirectly reflecting the current problem of excessive emissions from capture fisheries. Under the constraint of keeping the same total amount of resources before and after resource adjustment, the carbon emission resources of Jiangsu Seine net, Shanghai Drift net, Guangxi Fixed net, and Hebei Angling get increased, and the shadow price of carbon resources in these four sectors is relatively high, and the increased supply of resources can alleviate the carbon resource shortage in these four sectors. This paper does not set up a resource withdrawal mechanism, which leads to the concentration of carbon resources that should be withdrawn from allocation to these four sectoral industries.

**4.2.2 Carbon resource shadow price trend analysis.** Comparing the difference in shadow prices before and after optimization, most fishing methods show a downward trend. According to Fig 4, it can be seen that the algorithm shrinks the number of factors of production such as labor and fishing vessels, which makes the use of carbon resources become relatively abundant. The withdrawal of factors of production implies a reduction in the scale of fishing, which can be further considered in the context of the current overfishing situation. Because of the diminishing returns to scale theorem, it is beneficial to reduce fishing intensity. As fishing intensity decreases, the efficiency of carbon resource utilization increases, which means that

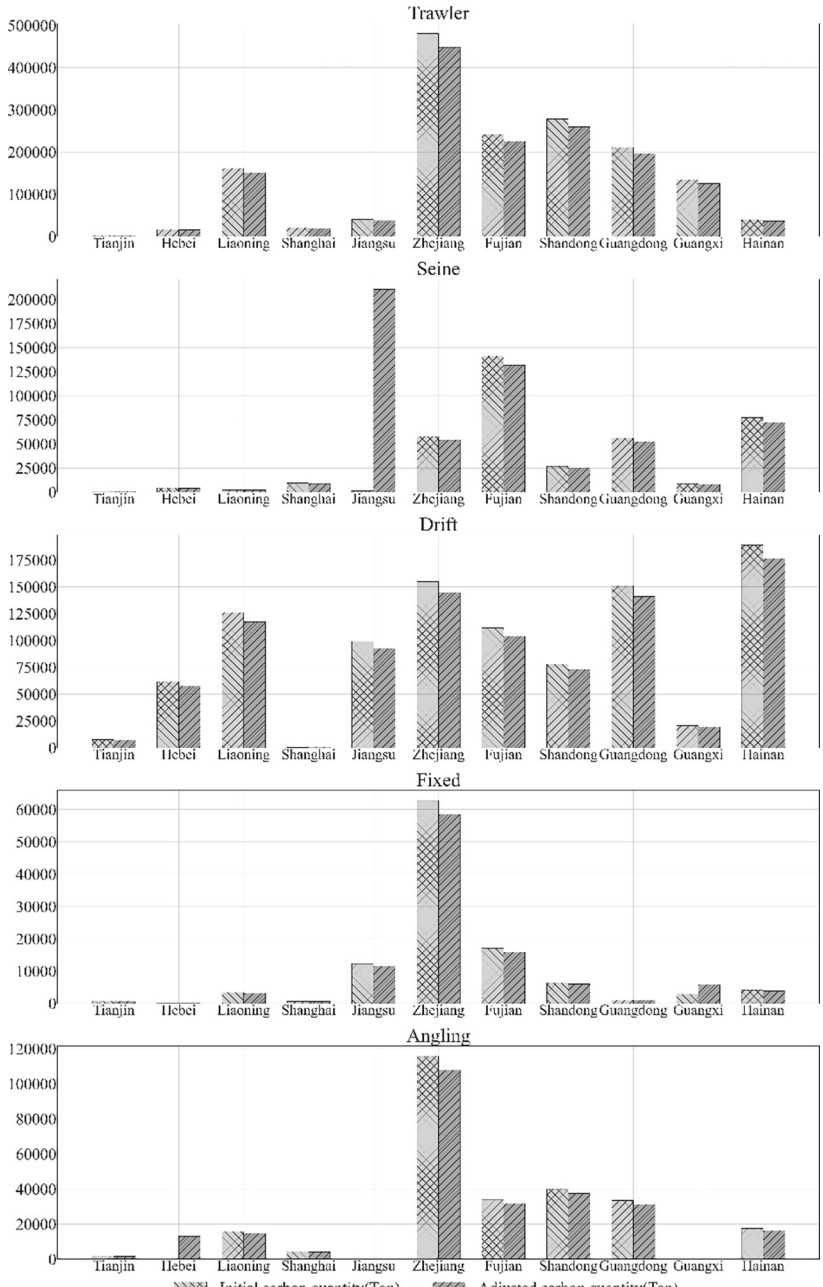

**Fig 4. Results of carbon resource allocation for 2000 iterations.**

relative abundance of carbon resources can be achieved, and this relative abundance to other factors is reflected in a smaller shadow price.

The fishing scheme with a shadow price of 0 before optimization was not changed during the optimization process, a situation that is mainly reflected in the Fixed net fishing method. Fixed net is a passive type of fishing, where the production process does not depend on fuel and uses as little fuel as possible, and therefore the algorithm did not correct for it on a large scale.

**Table 3. Shadow prices after optimal allocation.**

| Department | Trawler | Seine net | Drift net | Fixed net | Angling |
|---|---|---|---|---|---|
| Tianjin | 0.00497 | 0.01258 | 0.00172 | 0.01705 | 0.00763 |
| Hebei | 0.00735 | 0.02779 | 0.00198 | 0 | 0.35685 |
| Liaoning | 0.00052 | 0.03602 | 0.00067 | 0 | 0.0053 |
| Shanghai | 0.00296 | 0.00651 | 0.2488 | 0.09906 | 0.01404 |
| Jiangsu | 0.01387 | 0.39285 | 0.00567 | 0.0459 | 0 |
| Zhejiang | 0.00052 | 0.0043 | 0.00161 | 0.00397 | 0.00215 |
| Fujian | 0.0019 | 0.00324 | 0.0041 | 0 | 0.01354 |
| Shandong | 0.00217 | 0.02241 | 0.00772 | 0 | 0.01505 |
| Guangdong | 0 | 0.00744 | 0.00276 | 0 | 0.01249 |
| Guangxi | 0.00481 | 0.07379 | 0 | 0.22299 | 0 |
| Hainan | 0.00041 | 0.00021 | 0.00008 | 0 | 0 |

Table 3 shows the shadow price results after resource optimization. The values are compared with those in Table 2, and the comparison results are shown in Table 4. It is more meaningful to analyze changes in increase and decrease than to study changes in absolute numbers. Both the trawling and Angling industries are facing a state of shrinking carbon resources, but the reasons are not the same for both. Trawling is an industry with a high degree of fuel consumption, and the algorithm imposes a larger constraint on it. On the contrary, Seine net, being a green industry, the algorithm does not impose too many constraints, and this explanation can be drawn in conjunction with Fig 6. Trawling mainly reduces emissions through large-scale withdrawal of resources, while fishing reduces emissions through resource restructuring, so there is an essential difference in the adjustment principles between the two.

In Fig 4, the total amount of carbon emission resources of Tianjin Seine net is extremely small, which means that the scale reward of resources is in an increasing trend. When the total amount of resources is further reduced, it will inevitably cause irrational use of factors and cause an increase in shadow prices, so the increase of Tianjin Seine net can be accepted from Table 3.

**4.2.3 Regional analysis.** Tables 5 and 6 summarize the results of the available allocations and the optimized allocations in the regional dimension. These two tables show the changes in factors of production, and the algorithm uses the number of people employed in the fishery as an aggregate constraint, but in reality, the number of inputs to the labor force depends on the willingness of local residents to work. It is difficult to migrate the fishery labor force without considering changes in wage levels and incomes. Of course, this paper only solves the optimization solution for a fixed total amount of resources.

**Table 4. Increases and decreases in resources.**

| Department | Tianjin | Hebei | Liaoning | Shanghai | Jiangsu | Zhejiang | Fujian | Shandong | Guangdong | Guangxi | Hainan |
|---|---|---|---|---|---|---|---|---|---|---|---|
| Trawler | - | - | - | - | - | - | - | - | - | - | - |
| Seine net | + | - | - | - | - | - | - | - | - | - | - |
| Drift net | - | - | - | - | - | - | - | - | - | * | - |
| Fixed net | - | * | * | - | - | - | * | * | * | - | * |
| Angling | - | - | - | - | - | - | - | - | - | - | - |

Note:—indicates a decrease after optimization compared to 2018–2019; + indicates an increase after optimization compared to 2018–2019; * Indicates unchanged after optimization

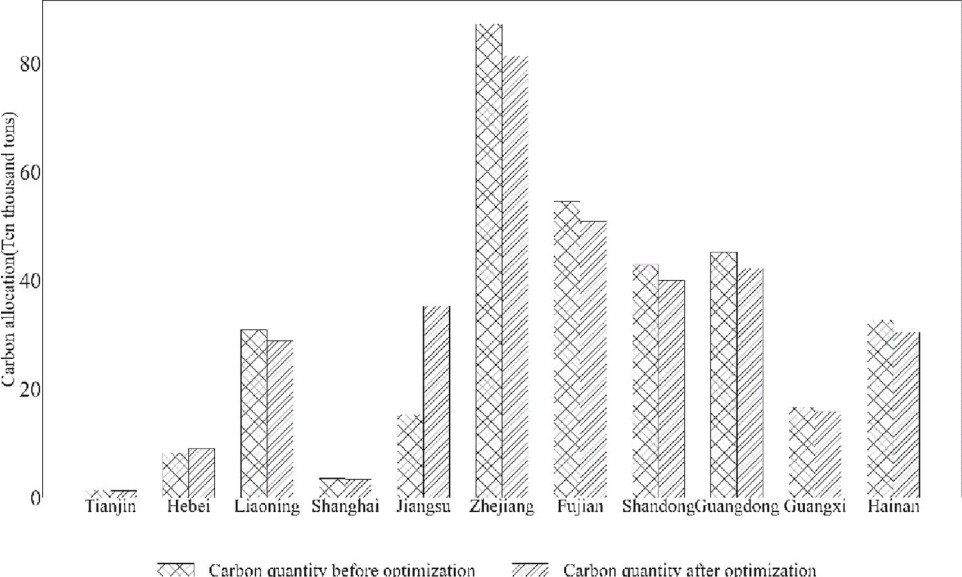

**Fig 5. Results of regional carbon resource optimization.**

Fig 5 shows the optimization changes of regional carbon resources. Of the 11 coastal provinces and cities, nine have suppressed their carbon emissions. In addition, the two provinces and cities with rising carbon emissions are Hebei Province and Jiangsu Province, with Jiangsu Province gaining the most carbon emissions. Zhejiang Province had more carbon emissions before the optimization, and did not have too much carbon emission constraints based on the initial higher share of carbon emission quantities. The second and third places are Fujian and Guangdong, respectively, as large provinces in the fishing industry, the optimization result appropriately reduces the total amount of carbon emissions.

In both Jiangsu and Hebei, the results show that more carbon compensation is given, and part of the compensation can be considered as compensation to the small-scale industry and the other part can be considered as resource withdrawal. The rationale is as follows: small-scale fisheries are more efficient in their use of carbon when resources are allocated. In the case of an overall oversupply of resources, the remaining resources are substantially compensated to the efficient small-scale sector according to the target fitness principle. However,

**Table 5. Initial factors of production.**

| District | Carbon emission (tons) | Production (tons) | Vessel displacement (tons) | Number of fishing vessels | Fishery worker |
|---|---|---|---|---|---|
| Tianjin | 13645.13871 | 32194 | 33032 | 410 | 117655 |
| Hebei | 82980.04812 | 159731 | 225368 | 3573 | 916425 |
| Liaoning | 309375.0838 | 451067 | 541196 | 14873 | 2633920 |
| Shanghai | 35216.62064 | 45742 | 94998 | 305 | 69475 |
| Jiangsu | 153519.1409 | 376165 | 326021 | 4806 | 5087790 |
| Zhejiang | 871893.2734 | 2572515 | 2345527 | 15317 | 3383050 |
| Fujian | 545353.4942 | 1436458 | 1089853 | 17477 | 4548395 |
| Shandong | 429275.5564 | 1634316 | 901140 | 18199 | 6570670 |
| Guangdong | 452453.1096 | 1150877 | 876984 | 31162 | 6194045 |
| Guangxi | 166710.8021 | 490953 | 457519 | 8996 | 4050180 |
| Hainan | 327101.7774 | 1024026 | 519060 | 23124 | 1221190 |

**Table 6. Factors of production after optimization.**

| District | Carbon emission (tons) | Production (tons) | Vessel displacement (tons) | Number of fishing vessels | Fishery worker |
|---|---|---|---|---|---|
| Tianjin | 12736.33726 | 30211.09239 | 29729.90662 | 383.7823294 | 19893.67122 |
| Hebei | 90145.61433 | 150170.1096 | 243843.6677 | 3493.25683 | 1385226.828 |
| Liaoning | 288517.8041 | 422774.1774 | 486588.4542 | 13915.79673 | 444978.7536 |
| Shanghai | 33367.47655 | 44059.4534 | 86362.5125 | 296.5272375 | 16636.5042 |
| Jiangsu | 352658.4925 | 931993.0353 | 991299.7828 | 11395.99542 | 27778401.13 |
| Zhejiang | 813236.4906 | 2411663.213 | 2109204.268 | 14334.38628 | 571363.4769 |
| Fujian | 508905.5701 | 1346964.174 | 980533.6926 | 16363.36836 | 768449.1231 |
| Shandong | 400719.9931 | 1532878.642 | 811053.5757 | 17045.57709 | 1110840.543 |
| Guangdong | 422225.1366 | 1079370.664 | 789057.3822 | 29174.96982 | 1046563.795 |
| Guangxi | 158800.7223 | 461212.69 | 412278.7458 | 10163.84468 | 842538.8214 |
| Hainan | 304982.2134 | 959609.9641 | 466565.6033 | 21630.55678 | 206108.2537 |

under fair emission reduction conditions, this strategy can constrain the role of carbon emissions. The optimized carbon emissions of Tianjin, Shanghai and Guangxi are approximately unchanged, and the scale of carbon emissions in these three places is initially relatively small, and the adjustment of carbon emission reduction is small.

**4.2.4 Analysis of fishing methods.** The estimation of fishing resource optimization is shown in Table 7. The Fig 6 indicates the carbon emission allowances for the two fishing types, Fixed net and Angling, are approximately constant after optimization. Trawling and Drift net

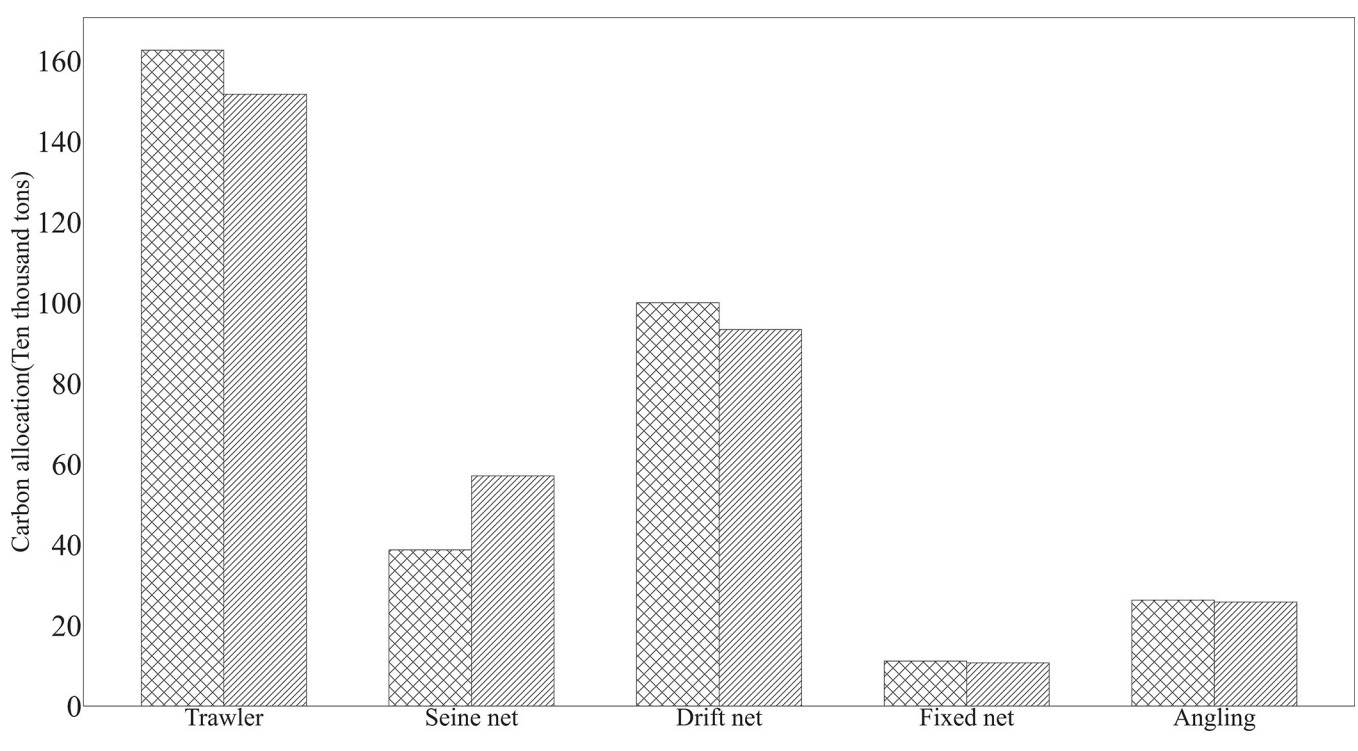

**Fig 6. Results of carbon resource optimization of fishing type.**

Table 7. The result of resource optimization by fishing type.

| Type | Before | | After | |
|---|---|---|---|---|
| | Carbon emission (tons) | Productive value(Ten thousand yuan,CNY) | Carbon emission (tons) | Productive value(Ten thousand yuan,CNY) |
| Trawler | 1626111.988 | 8305349.597 | 1516887.811 | 6922276.907 |
| Seine net | 387195.444 | 2514353.954 | 570741.4969 | 5263857.533 |
| Drift net | 1000094.529 | 5576491.151 | 933536.7978 | 4650769.706 |
| Fixed net | 111434.8731 | 1038939.728 | 107136.1541 | 867212.7199 |
| Angling | 262687.2113 | 2218726.289 | 257993.5909 | 1930241.055 |

are optimized to a greater extent, shrinking a large number of carbon emission quotas, while Seine net receives a larger allocation of carbon quotas, as seen in Tab.7 and Fig 6. Analyzed from a practical point of view, Seine net is an efficient fishing method compared to trawl, and the fishing method is more targeted, thus its fishing efficiency is higher. The fishing method of gillnets needs to rely on the movement of fish and it is difficult to predict the possible activity areas of fish, therefore, the fishing method of Seine net is more targeted and its carbon emission benefit is relatively higher. As a result, Seine net should be compensated with more carbon resources. The lower energy consumption of Angling is matched by the low output value of Angling fishing operations, which results in no particular advantage in terms of costs and benefits for Angling. Fixed net are also a passive form of fishing that is used on a smaller scale because of their lower capacity.

## 5 Further discussion

Differences in sensitivity to carbon emissions exist between capture fisheries. When the pressure to reduce emissions of the same scale intensity is applied to different industries, there is a difference in the potential output of carbon resources, and this difference not only makes the carbon reduction effect much less effective, but also raises the issue of income inequity. With carbon resources as a cost constraint, this inequity in carbon resource factors can lead actors to choose factor flows or exit. Therefore, there is a need to develop adapted abatement programs for different industries. In this paper, the potential output consistency of carbon resources is used as the equity criterion for carbon factor resources.

Discussed in economic terms, the shadow price is a marginal price where input factors can be interpreted in a marginal way. When the shadow prices are the same, i.e., the marginal factor costs of carbon resource factors remain the same, it means that the factors in each industry are in a steady state and there is no formation of underground markets or factor shifts. Further analyzing the results, this turns out to be a stable equilibrium point for carbon resource factors [49], i.e., there will be no inflow of resources due to high profitability of carbon factors in a particular industry, and there will be no outflow of carbon resources due to low profitability of carbon resource factors in a particular industry. This stabilization point is considered to be short-term stabilization [50]. However, in the long run, the industry can still adjust other factors of production to improve production efficiency, and when the factor ratios change, for example, the amount of capital or labor factor inputs can be changed through technological factor inputs [51], the carbon resource factor will be adjusted. This paper does not set the corresponding technical progress function curve, and further research in the future will take more constraints and limitations into account.

A more efficient use of carbon resources means that more output can be produced per unit of carbon resources. Conversely, a less efficient use of carbon resources means that less output can be produced per unit of carbon resources. When a uniform carbon reduction intensity is

applied to different industries, when the industries with higher and lower carbon resource utilization efficiency reduce the same unit of carbon resources, the number of outputs lost by the enterprises with higher carbon resource utilization efficiency is much larger than that of the enterprises with lower carbon resource utilization efficiency.

This implies that in order to maintain the same revenue from carbon resource utilization, more efficient carbon resource utilization industries can downsize a smaller amount of carbon resources, and the revenue loss they reduce is the same as the revenue loss of less efficient carbon resource utilization industries that reduce a larger amount of carbon resources. Therefore, the industry with higher efficiency of carbon resources utilization can reduce less amount of carbon resources, while the industry with lower efficiency of carbon resources utilization can reduce more amount of carbon resources. When the shadow prices of the two are the same, it means that the industry with higher carbon resource utilization efficiency and the industry with lower carbon resource utilization efficiency will keep the output value per unit of carbon resource at the same level.

## 6 Summary and outlook

China's Marine capture carbon resources are faced with the problem of unreasonable distribution, high carbon emission operation mode of carbon resources are sufficient, low carbon emission operation mode of carbon resources supply is short. The excessive carbon emissions of capture fisheries and the expansion of production scale by using the advantages of carbon resources in some areas will lead to the tragedy of the public land and the decline of Marine fishery resources.

This paper measures the shadow price of carbon resources as the basis for resource allocation. The author argues that the equity of resource management requires that different fishing methods in different regions should obtain the same economic output for each additional unit of carbon resource input. Therefore, this study takes the consistency of the shadow price of carbon resources as the value assessment basis for assuming the responsibility of fair governance, and uses the difference between the individual value and the ideal target value to correct the unreasonable distribution of carbon resources in the current Marine capture fisheries. After resource adjustment, the problem of resource redundancy or shortage caused by unreasonable allocation of carbon resources is solved. The conclusions of this study are as follows.

First, from a regional perspective, carbon emission resources are allocated to the large fishing provinces of Hebei and Jiangsu, while the total carbon resources of Liaoning, Zhejiang, Fujian, Shandong, Guangdong, Guangxi and Hainan are reduced. For the two municipalities directly under the central government, Shanghai and Tianjin, the total amount of carbon resources is basically unchanged. The fishery products in these two places have a large market demand, mature consumer market, production technology can get more capital boost, and the proportion of production factors is more reasonable, so the two can be in a constant state.

Second, analyzing from the perspective of fishing operation mode, the energy-saving and emission-reducing seine net gained a larger carbon allocation; the carbon resources of trawl and drift net were scaled down; and the fixed net and angling remained unchanged, which were smaller in scale while their carbon resource allocation was more flexible and reasonable. Trawling has been the fishing method with the larger fuel carbon emissions. Seine net requires more skillful and collaborative fishing than trawling, which produces fish by accurately detecting the location of fish. Algorithms give seine nets more carbon resources. Similarly, fixed net and angling are smoother, their fishing methods are more environmentally friendly, and although they are not as economically efficient, their carbon resource emission efficiency is

more stable, and the optimization did not result in a change in the total amount of carbon resources.

Thirdly, the allocation of carbon resources should be took into account for the difference between passive and active fishing. Active fishing is widely used by fishermen because of its high economic returns, but active fishing is not took into account for the cost of carbon emissions, which leads to fishermen's willingness to increase their income through excessive carbon emissions. Therefore, for active fishing methods, the cost of carbon resources should be reflected in the production process, so the trawl scale should be limited and the seine net scale should be increased appropriately in active fishing production. Passive fishing does not rely heavily on carbon resources, so more emphasis should be placed on economically efficient fishing methods in order to increase the scale of fishery production for clean fishing.

The results of the shadow price analysis of carbon resources of capture fisheries show that: the current carbon resources of marine capture fisheries face the problem of unreasonable allocation, i.e., the carbon resources of high carbon emission operation methods are sufficient and the carbon resources of low carbon emission operation methods are in short supply. The carbon emissions of capture fisheries are too large, and the expansion of production scale in some areas by taking advantage of carbon resources will lead to the phenomenon of tragedy of the commons and cause the decline of marine fishery resources. This model uses the consistency of the shadow price of carbon resources as the basis for judging the value target of assuming the responsibility of fair governance, and corrects the unreasonable problem of carbon resource allocation in the current marine capture fisheries by using the difference between the individual value and the ideal target value. After resource adjustment, the problem of redundancy or shortage of resources caused by unreasonable allocation of carbon resources in regions and operation types is solved. To simplify the model, it is assumed that the technology level of production sectors in each region does not change, which leads to the input-output ratios of resources in the optimization process being set according to the sectoral input-output ratios in the previous period, which is not the case in reality. That is, the production sectors with large amounts of resources will change their input-output ratios due to scale effects or technological progress. Therefore, the resource allocation model with technological progress can be further discussed in the future.

## Author Contributions

**Conceptualization:** Guangliang Li.

**Funding acquisition:** Hailan Qiu, Chunlan Tan.

**Investigation:** Weikun Zhang, Juanjuan Niu.

**Project administration:** Hailan Qiu.

**Writing – original draft:** Guangliang Li.

**Writing – review & editing:** Guangliang Li.

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
