## [Decision Letter · Decision Letter 0]

17 Jul 2023

PONE-D-22-30576Adjustment of carbon resource allocation in marine capture fisheries in the context of equity —— A new allocation method based on shadow pricesPLOS ONE

Dear Dr. Li,

Thank you for submitting your manuscript to PLOS ONE. After careful consideration, we feel that it has merit but does not fully meet PLOS ONE’s publication criteria as it currently stands. Therefore, we invite you to submit a revised version of the manuscript that addresses the points raised during the review process.

We look forward to receiving your revised manuscript.

Kind regards,

Chaohai Shen

Academic Editor

PLOS ONE

Journal Requirements:

2. Please ensure that you include a title page within your main document. You should list all authors and all affiliations as per our author instructions and clearly indicate the corresponding author.

Reviewers' comments:

Reviewer's Responses to Questions

**Comments to the Author**

1. Is the manuscript technically sound, and do the data support the conclusions?

Reviewer #1: Partly

Reviewer #2: Yes

Reviewer #3: Yes

2. Has the statistical analysis been performed appropriately and rigorously? 

Reviewer #1: I Don't Know

Reviewer #2: Yes

Reviewer #3: Yes

3. Have the authors made all data underlying the findings in their manuscript fully available?

Reviewer #1: Yes

Reviewer #2: Yes

Reviewer #3: Yes

4. Is the manuscript presented in an intelligible fashion and written in standard English?

Reviewer #1: No

Reviewer #2: Yes

Reviewer #3: Yes

5. Review Comments to the Author

Reviewer #1: This paper addresses an important issue. The authors mainly presented their own interpretation of the findings without referring to any literature. How this innovation would’ helps to promote the matching of social values and resource utilization, corrects the profit-oriented social contradiction,

and provides new ideas for the study of industrial layout and optimization of resource utilization in research and application in economics’ should further be critically analyzed with support of scientific literature.

Reviewer #2: This paper, titled as “Adjustment of carbon resource allocation in marine capture fisheries in the context of

equity”, aims to r takes social responsibility equity as the value goal orientation and constructs a resource allocation model to correct the unfair distribution of carbon resources in fishery, most of which have been constructed in the new town area by the provical government to alleviate the development pressure to carbon resources or industrial structure, focusing on the environment friendly green fishing mode. The topic is important and the methodologies used for analysis is quite interesting.

However, the paper needs to clarify the following points, and major revision.

The definition of main objects of this research should be clearly shown;

Working hypothesis of this paper may need to be carefully explained. I agree the following conclusion

However, I hesitate to endorse the conclusion that A new allocation method based on shadow prices and contributed to increase of carbon resource allocation , due to lack of backup data in this paper

Reviewer #3: 1. Authors should sort out the abstract by stating the importance of the research question, the content of the study, and the novel conclusions and insights without repetitively summarizing ``Innovation’’.

2. In the introduction, I did not clearly see the importance and urgency of the research problem, the knowledge gaps with previous research, the challenges that still need to be addressed urgently, and authors’ contributions that have been made to address the challenges.

3. The review of relevant literature in this manuscript is very lacking and inadequate.

4. Figure 1 summarizes the decision-making process. What is the difference between this and the relevant literature?

5. Technical variables should depict the current technical level and ability, but can the number of technology upgrading institutions and technology upgrading funds truly reflect the technical level? Are these two indicators positively correlated with technical level?

6. The manuscript should be written in a more standardized way, especially the font of the text and the body of the figures should be consistent.

7. Comparing before and after resource adjustment, are the change in carbon resources for Seine net sector in Shanghai in line with reality?

8. Both the experiment in Section 4 and the corresponding analysis of the results were inadequate.

9. In the conclusion, the author should state the unique conclusions and put forward relevant insights and suggestions. Also, the author should state future research based on the main content.

6. PLOS authors have the option to publish the peer review history of their article (what does this mean?). If published, this will include your full peer review and any attached files.

Reviewer #1: No

Reviewer #2: No

Reviewer #3: No

---

## [Author Response · Author response to Decision Letter 0]

31 Aug 2023

Reviewer 1's comments

This paper addresses an important issue. The authors mainly presented their own interpretation of the findings without referring to any literature. How this innovation would’ helps to promote the matching of social values and resource utilization, corrects the profit-oriented social contradiction, and provides new ideas for the study of industrial layout and optimization of resource utilization in research and application in economics’ should further be critically analyzed with support of scientific literature.

Thank you very much for reviewing this article. On behalf of all the authors, I would like to thank you.

Reviewer 1#Reply1

This method was inspired after encountering the computational power limitation of DEA algorithm. Resource allocation is in pursuit of a particular goal, which adjustment process is extremely similar to the training of neural networks. Therefore, this paper adopts some of the ideas of neural network calculation process to assess the change of the result through the small change of the input, and finally gradually approximates the ideal value through the adjustment of the input elements.(Line 145)

The innovation is that the steps of resource allocation in this paper are divided into four major steps, value assessment, optimization objective, difference between value assessment and objective, and optimization calculation. This allocation step is also easier to understand.(line 135-144)

At the same time, this article has made the following changes:

The author has made several adjustments to the subsequent analysis, including the addition of "5 Further discussion" section and Figures 4 to 6. These figures provide a detailed comparison of the changes before and after the allocation of carbon resources. The action mechanism of the algorithm is explained in Figure 1, and the main idea is thoroughly discussed in section "2 Theory". The method model primarily employs an iterative approach to continuously revise towards the desired goal. The economic implications of the results are discussed in section "5 Further discussion". Additionally, the future prospects for in-depth research are outlined in section "6 Summary and outlook".

Location of the corresponding changes in the manuscript: line 378-675

Reviewer 1#Reply2

The author has added relevant references. This paper draws on some content of neural network algorithm, but not all of it. The derivation is given in the paper, and relevant references are added in the derivation part.

Location of the corresponding changes in the manuscript: line 232;236;244；245;247;255

Reviewer 2's comments

This paper, titled as “Adjustment of carbon resource allocation in marine capture fisheries in the context of equity”, aims to r takes social responsibility equity as the value goal orientation and constructs a resource allocation model to correct the unfair distribution of carbon resources in fishery, most of which have been constructed in the new town area by the provical government to alleviate the development pressure to carbon resources or industrial structure, focusing on the environment friendly green fishing mode. 

The topic is important and the methodologies used for analysis is quite interesting.

However, the paper needs to clarify the following points, and major revision.

The definition of main objects of this research should be clearly shown;Working hypothesis of this paper may need to be carefully explained.

 I agree the following conclusion.However, I hesitate to endorse the conclusion that A new allocation method based on shadow prices and contributed to increase of carbon resource allocation , due to lack of backup data in this paper

Thank you very much for reviewing this article. On behalf of all the authors, I would like to thank you.

Reviewer 2#Reply1

This paper has been improved according to the reviewer's opinions, adding a lot of content, and adding charts and explanations for this purpose.

The research objectives of this paper are explained in lines 72-78. This study aims to achieve equitable and efficient carbon resource allocation for marine capture fisheries. (line 72-78)

In Section 5, we discuss the significance and rationality of our research conclusions. We assume that technological progress is negligible, which leads to a short-term equilibrium rather than a long-term one.(line 558-602)

Reviewer 2#Reply2

The data in this paper are supported by a clear explanation, and the data are cited.(line 192-194)

The conclusion is consistent with the real policies. China's coastal provinces have stopped fuel subsidies for trawlers and promoted cleaner production methods. Specific facts can be found at:

From 2020, the subsidy will no longer be given to fishing vessels with double wing and single bag trawling (double bottom trawling), single anchorage and single bag purse Seine.

Link:

https://zhuanlan.zhihu.com/p/64791738

https://www.gov.cn/xinwen/2015-07/09/content_2894870.htm

https://www.gdzwfw.gov.cn/portal/v2/affairs-public-detail?qzqdCode=A924E8C7F7761194E0530C3D10ACF992&deptCode=441024868

Reviewer 3's comments

1. Authors should sort out the abstract by stating the importance of the research question, the content of the study, and the novel conclusions and insights without repetitively summarizing ``Innovation”.

2. In the introduction, I did not clearly see the importance and urgency of the research problem, the knowledge gaps with previous research, the challenges that still need to be addressed urgently, and authors’ contributions that have been made to address the challenges.

3. The review of relevant literature in this manuscript is very lacking and inadequate.

4. Figure 1 summarizes the decision-making process. What is the difference between this and the relevant literature?

5. Technical variables should depict the current technical level and ability, but can the number of technology upgrading institutions and technology upgrading funds truly reflect the technical level? Are these two indicators positively correlated with technical level?

6. The manuscript should be written in a more standardized way, especially the font of the text and the body of the figures should be consistent.

7. Comparing before and after resource adjustment, are the change in carbon resources for Seine net sector in Shanghai in line with reality?

8. Both the experiment in Section 4 and the corresponding analysis of the results were inadequate.

9. In the conclusion, the author should state the unique conclusions and put forward relevant insights and suggestions. Also, the author should state future research based on the main content.

Thank you very much for reviewing this article. On behalf of all the authors, I would like to thank you.

Reviewer 3#Reply1

The abstract has been revised.

Marine fishery carbon emissions play a significant role in agricultural carbon emissions, making resource allocation a crucial topic for the overall marine ecological protection. This paper evaluates the dynamic iteration method as a research approach with the factors of resource allocation consisting of value assessment, optimization objective, difference between value assessment and objective, and optimization calculation. The paper selects the shadow price from the Super-SBM model as the judgment function for the goal value, aiming for the fairness criterion. From an equity standpoint, the allocation of carbon resources in marine capture fisheries proves to be unreasonable. The fishery model exhibits an excessive supply of carbon resources, resulting in wastage, while the green fishery model faces a relatively limited supply, with a focus on energy conservation and environmental protection. To address this issue, this paper proposes a new method and discusses the corrective results. This result shows that the stabilization point achieved is a short-term equilibrium rather than a long-term one. By rectifying the social contradiction of profit-oriented approaches, this research provides a fresh perspective for economic studies and applications, particularly in industrial layout and resource utilization optimization.(line 24-39)

Reviewer 3#Reply2

Marine carbon emission reduction has always been an important research topic. In the introduction, we revisit this topic and outline the following contents: the background of the problem, the related research on unfair resource allocation, the existing research on carbon resource allocation by scholars, and the discussion of the existing research and our new design scheme in this paper. (line 46-110)

Reviewer 3#Reply3

This article rewrites the introduction.(line 46-110)

Reviewer 3#Reply4

Figure 1 shows the steps for resource allocation. The innovation of this paper is that it divides the resource allocation process into four major steps: value assessment, optimization objective, difference between value assessment and objective, and optimization calculation. This method is clearer and more systematic than other studies of the same type.

This step follows the decision-making process, which consists of setting goals, evaluating oneself, measuring the gap between oneself and the goals, and implementing improvement plans.(line 135-150)

Reviewer 3#Reply5

Technology promotion agencies are essential for the production process, as they can facilitate the technological upgrading of the production end. This paper considers patents as a measure of technology, but patents are only theoretical and need to be transformed into social production by technology promotion organizations.

Reviewer 3#Reply6

The author has made corrections.

Reviewer 3#Reply7

Figures 4, 5, and 6 show the comparison before and after optimization. The purse seine work in Shanghai is consistent with the realistic conditions. This paper discusses and analyzes Shanghai and Tianjin.

For the two municipalities directly under the central government, Shanghai and Tianjin have similar amounts of carbon resources. The fishery products in these two places have high market demand, mature consumer markets, and advanced production technologies. The proportion of production factors is more reasonable, so the two can maintain a constant state.(line 623-628)

Reviewer 3#Reply8

A lot of content has been added and significant changes have been made to the analysis section. (line 378-602)

Reviewer 3#Reply9

Thanks very much for the reviewer's comments, this part has been significantly modified.(line 604-672)

---

## [Decision Letter · Decision Letter 1]

2 Oct 2023

PONE-D-22-30576R1The allocation of carbon resources in marine capture fisheriesPLOS ONE

Dear Dr. Qiu,

Thank you for submitting your manuscript to PLOS ONE. After careful consideration, we feel that it has merit but does not fully meet PLOS ONE’s publication criteria as it currently stands. Therefore, we invite you to submit a revised version of the manuscript that addresses the points raised during the review process.

We look forward to receiving your revised manuscript.

Kind regards,

Chaohai Shen

Academic Editor

PLOS ONE

Journal Requirements:

Additional Editor Comments (if provided):

Dear Authors,

Please address the comments from one reviewer. Thanks.

Best,

Reviewers' comments:

Reviewer's Responses to Questions

**Comments to the Author**

1. If the authors have adequately addressed your comments raised in a previous round of review and you feel that this manuscript is now acceptable for publication, you may indicate that here to bypass the “Comments to the Author” section, enter your conflict of interest statement in the “Confidential to Editor” section, and submit your "Accept" recommendation.

Reviewer #1: All comments have been addressed

Reviewer #2: (No Response)

Reviewer #3: (No Response)

2. Is the manuscript technically sound, and do the data support the conclusions?

Reviewer #1: Yes

Reviewer #2: Yes

Reviewer #3: (No Response)

3. Has the statistical analysis been performed appropriately and rigorously? 

Reviewer #1: I Don't Know

Reviewer #2: Yes

Reviewer #3: (No Response)

4. Have the authors made all data underlying the findings in their manuscript fully available?

Reviewer #1: Yes

Reviewer #2: Yes

Reviewer #3: (No Response)

5. Is the manuscript presented in an intelligible fashion and written in standard English?

Reviewer #1: Yes

Reviewer #2: Yes

Reviewer #3: (No Response)

6. Review Comments to the Author

Reviewer #1: (No Response)

Reviewer #2: All comments have been revised. No new issues were found, and it is recommended that the journal be accepted .

Reviewer #3: The author has made most of the modifications according to my opinion, but the presentation of Figure 4 is very unclear, which will not only devour the author's contribution and work but also affect the readers' perception of the whole article. The author should improve this figure.

Moreover, the font size in the figure and part of the table is inconsistent, and some are too small, which affects the perception.

In addition, the current presentation of Table 2 and Table 4 does not fit the margins, and the author should change the expression to make it more compliant with the specification.

7. PLOS authors have the option to publish the peer review history of their article (what does this mean?). If published, this will include your full peer review and any attached files.

Reviewer #1: No

Reviewer #2: No

Reviewer #3: No

---

## [Author Response · Author response to Decision Letter 1]

5 Oct 2023

Reviewer1

Thank you for reviewing the manuscript.

Reviewer2

Thank you for reviewing the manuscript.

Reviewer3

Thank you very much for your comments. The paper has adjusted the tables2,tables4 and Figure 4.

---

## [Editor Report · Decision Letter 2]

6 Oct 2023

The allocation of carbon resources in marine capture fisheries

PONE-D-22-30576R2

Dear Dr. Qiu,

We’re pleased to inform you that your manuscript has been judged scientifically suitable for publication and will be formally accepted for publication once it meets all outstanding technical requirements.

Kind regards,

Chaohai Shen

Academic Editor

PLOS ONE
---

## [Editor Report · Acceptance letter]

6 Mar 2024

PONE-D-22-30576R2 

PLOS ONE

Dear Dr. Qiu, 

I'm pleased to inform you that your manuscript has been deemed suitable for publication in PLOS ONE. Congratulations! Your manuscript is now being handed over to our production team.

Kind regards, 

on behalf of

Dr. Chaohai Shen 

Academic Editor

PLOS ONE